# Berberine and Its Study as an Antidiabetic Compound

**DOI:** 10.3390/biology12070973

**Published:** 2023-07-08

**Authors:** Ayudiah Rizki Utami, Iman Permana Maksum, Yusi Deawati

**Affiliations:** Department of Chemistry, Faculty of Mathematics and Natural Sciences, Universitas Padjadjaran, Sumedang 45363, Indonesia; ayudiah19001@mail.unpad.ac.id (A.R.U.); yusi.deawati@unpad.ac.id (Y.D.)

**Keywords:** diabetes mellitus, hyperglycemia, AMPK pathway, antidiabetic compound

## Abstract

**Simple Summary:**

Glucose is needed as the main source of energy in the body. However, too high sugar levels can lead to a variety of serious complications, including with the cardiovascular system, eyesight, and kidney function. The condition in which blood sugar levels are high is called hyperglycemia. Diabetes mellitus is a metabolic disease that causes hyperglycemia and is a disease that has been known to cause many deaths. Therefore, many treatments and preventive measures have been taken to fight diabetes mellitus. There are several oral diabetic drugs used, such as metformin, sulfonylureas, rosiglitazone, and others. However, some of these drugs have high risks or have contraindications with certain groups. Here, we discussed the mechanism of the natural compound berberine as an antidiabetic compound. In addition, we also found that berberine can work as an anti-inflammatory and antioxidant, where the state of inflammation and oxidative stress itself has an influence on the condition of diabetes mellitus.

**Abstract:**

Diabetes mellitus (DM) is a metabolic disorder that causes hyperglycemia conditions and leads to various chronic complications that causes death. The prevalence of diabetes is predicted to continue to increase, and with the high toxicity levels of current diabetes drugs, the exploration of natural compounds as alternative diabetes treatment has been widely carried out, one of which is berberine. Berberine and several other alkaloid compounds, including some of its derivatives, have shown many bioactivities, such as neuraminidase and hepatoprotective activity. Berberine also exhibits antidiabetic activity. As an antidiabetic compound, berberine is known to reduce blood glucose levels, increase insulin secretion, and weaken glucose tolerance and insulin resistance by activating the AMPK pathway. Apart from being an antidiabetic compound, berberine also exhibits various other activities such as being anti-adipogenic, anti-hyperlipidemic, anti-inflammatory, and antioxidant. Many studies have been conducted on berberine, but its exact mechanism still needs to be clarified and requires further investigation. This review will discuss berberine and its mechanism as a natural compound with various activities, mainly as an antidiabetic.

## 1. Introduction

Diabetes mellitus (DM) is a metabolic disease where the patients have high blood sugar levels conditions (commonly called hyperglycemia) caused by metabolic disorders of carbohydrates, fats, and proteins in the body. Patients with DM are characterized by several symptoms, such as weight loss, difficult-to-heal wounds, persistent loose stools, skin problems, and visual impairment. DM can cause various complications in organs, such as damage in blood vessels, eye damage (diabetic retinopathy), kidney failure (diabetic neuropathy), visual impairment/blindness (diabetic retinopathy), nerve disorders (diabetic neuropathy), and heart attack and stroke (cardiovascular disease) [1]. Based on data taken from the International Diabetes Federation in 2021 [2], around 537 million adults in the world in the range of 20–79 years have been diagnosed with diabetes, with the prevalence of diabetes predicted to increase to 643 million by 2030 and 783 million by 2045. Until now, it has been recorded that diabetes has caused up to 6.7 million deaths per year.

DM can be categorized into two types, namely type 1 DM and type 2 DM. T1DM is a type of DM that occurs due to damage to the β-pancreatic cells responsible for insulin production and secretion. Hence, the level of insulin the body needs becomes less or even absent. This condition is known as insulin deficiency. The lack of insulin levels in T1DM patients leads to the need for insulin injection as a treatment method. Therefore, T1DM is also known as Insulin-Dependent Diabetes Mellitus (IDDM). T1DM can be found at a young age, such as in children and adolescents. T2DM, on the other hand, is a metabolic disease that causes insulin resistance. Insulin resistance is the inability of the body’s cells to respond to insulin hormone signals, resulting in impaired glucose metabolism. This condition occurs due to the influence of unhealthy lifestyles such as food, lack of exercise, obesity, increased high-fat diets and lack of fiber, age, and genetic factors [3]. In addition, the accumulation of fatty acids such as diacylglycerol (DAG) and ceramides can also cause insulin resistance [4]. T2DM usually appears at the age of over 30 years, but T2DM has also begun to be found at a young age in recent years [1,5].

There is another type of diabetes mellitus called mitochondrial diabetes which is a part of T2DM. Mitochondria are cytoplasmic organelles where cellular respiration occurs, and mitochondria primarily produce Adenosine Triphosphate (ATP) as chemical energy. Unlike nuclear DNA, mitochondrial DNA (mtDNA) has a high mutation rate due to the absence of a DNA repair mechanism [6,7].

Mitochondrial diabetes mainly occurs due to mutations in mitochondrial DNA (mtDNA), thus why it is called mitochondrial diabetes. These mutations are then associated with reduced function of pancreatic β-cell insulin. One of the mutations found is the mutation of the A3243G nucleotide of the tRNA^Leu^ mtDNA gene, which is heteroplasmy that causes maternally inherited diabetes accompanied by hearing loss [6,8,9]. Some of the other mutations that are associated with diabetes are C3271T, C12258A, A8296G, which causes cardiomyopathy [10], and T4291T, which is associated with myopathy, hypomagnesemia, and hypokalemia [11]. There is also a G9053A mutation at the ATP6 gene that was found in T2DM patients with cataracts, and its assay detection research was conducted by Ilmi et al. in 2023 along with the T15663C mutation in the CYB gene [12,13]. It is known that a mutation of mtDNA is responsible for approximately 80% of adult mitochondrial diseases [7]. One of the effects of mutations in mitochondrial functions results in a lack of ATP that causes dysfunction in the secretion of insulin [14].

DM, as one of the diseases that causes widespread mortality, has been widely researched, and its treatment has been made. The main method of treating T1DM patients is through insulin injection. Meanwhile, the treatment of T2DM generally uses the same approach for both children and adults. Overall, treatments for T2DM patients include weight loss, increased exercise routine, normalization of plasma glucose levels, and control of comorbidities, including hypertension, cardiomyopathy, nephropathy, and hepatic steatosis [15].

Various drugs have been used to treat T2DM, including insulin, acarbose, rosiglitazone, metformin, and sulfonylureas. These drugs can cause various side effects, including gastrointestinal symptoms [16]. Only a few of these drugs are approved for consumption by children. Insulin and metformin, which are approved drugs, are unable to achieve optimal glycemic control in most adolescents and pose a high risk to later health [17]. In addition, clinical trials show that metformin is insufficient to control comorbidities in T2DM patients, where 33.8% of patients show hypertension, 16.6% show microalbuminuria, and 13.7% show retinopathy [18]. On the other hand, insulin has a high risk of significant weight gain and hyperglycemia. There are some drugs that should only be used in certain groups of patients, such as metformin and rosiglitazone, which are contraindicated in patients with renal impairment, hepatic disease, heart failure, or respiratory problems [19].

The high risk posed by these antidiabetic drugs encourages the need to explore drugs for DM treatment. One of the measures that can be taken is the modification of the structure of the substrate. An example is the modification of iminosugar structures such as glucosidase inhibitors so that they mimic substrate conformation and/or positively charged transition states [20,21]. These structurally modified sugars represent targets for the development of antidiabetic activity [22]. In addition to structural modification, there is also an exploration of natural compounds as an alternative treatment for diabetes. According to Kesuma et al. (2018), bioactive compounds from natural compounds can be used as an alternative treatment for T2DM due to their low toxicity compared to synthetic drugs [23]. One of the natural compounds that has been explored is the berberine compound. Berberine is known to work against diabetes through several effects and mechanisms [19,24]. Berberine is comparable to other common diabetes treatments, such as metformin and sulfonylureas, which are used as first-line medications for treating type 2 diabetes in terms of their ability to lower glucose levels [25]. Berberine is also said to improve fat metabolism [25,26]. Although many studies have been conducted on berberine, few have revealed the exact mechanism of action. This article will review the research progress on berberine and its mechanism as an antidiabetic compound.

## 2. Berberine as a Therapeutic Natural Compound

Alkaloid compounds have been widely used in traditional medicine. Alkaloid compounds have various pharmacological activities, including antimalarial, antihyperglycemic, anti-asthma, anticancer, antibacterial, and antidiabetic attributes. One of those alkaloid compounds named berberine is an isoquinoline alkaloid that acts as a therapeutic agent in the treatment of T2DM [19,27,28]. Berberine has been used as a traditional medicine in China, India, and the Middle East region for more than 400 years [19,27]. Berberine has a quarternary base structure that can be seen in Figure 1, which makes it different from other hypoglycemic agents such as sulfonylureas, biguanides, thiazolidinediones, or acarbose [19]. Berberine is known to reduce blood glucose levels, increase insulin secretion, reduce body weight and lipid levels, attenuate glucose tolerance and insulin resistance by activating the 5′-adenosine monophosphate-activated protein kinase (AMPK) pathway, increase glucagon-like peptide-1 (GLP-1) levels, attenuate reactive oxygen species (ROS) production, reverse mitochondrial dysfunction, and suppress inflammation [29].

One of the studies conducted by Di et al. (2021) [30] showed that berberine, as a therapeutic antidiabetic agent, can alleviate diabetes and its complications through several targets and signaling pathways. A molecular docking study by Mohanty et al. (2017) [31] showed that berberine has a smaller binding affinity value to dipeptidyl peptidase-IV (DPP-IV) than the standard vildagliptin. On the other hand, research conducted by Mandar et al. (2021) [32] showed that berberine has a lower binding affinity value to α-amylase when compared to its standard inhibitor, acarbose. From these studies results, it can be surmised that berberine is less favorable in enzymatic antidiabetic treatment compared to its standard drug. 

Structurally, berberine belongs to the benzylisoquinoline group. Besides the genus *Berberis*, berberine can be isolated from another genus, such as *Coptis*, Corydalis, or *Mahonia.* Compounds a–g (Figure 2) were isolated from the roots of *Corydalis turtschaninovii*. These compounds showed strong neuraminidase inhibition activity with IC50 of 12.8–65.2 μM, which can be seen in Table 1 [33]. The function of neuraminidase is to hydrolyze the bond between the host cell receptor and the hemagglutinin of the influenza virus in the process of releasing new virus cells that have been formed from the virus replication process in the host cell. Thus, inhibition of this neuraminidase can prevent the release of the virus, which will then reduce its pathogenicity [33,34,35]. In addition, several berberine derivatives show hepatoprotective activity, including *dehydroisoapocavidine* (h) and *dehydrocheilanthifoline* (i), which can be isolated from *C. tomentella* [33].

## 3. Berberine and Structure–Activity Relationship (SAR)

Berberine is one of the natural compounds that has been used as traditional medicine in various regions of the world and is categorized as an alkaloid group compound, precisely an isoquinoline alkaloid. Berberine is known for its activity as an antidiabetic agent. One of the other alkaloid compounds known to have antidiabetic effects is coptisin. Coptisin is a compound of the isoquinoline alkaloid group that is known for its bitter taste and is widely used in herbal medicine in China to treat digestive disorders caused by bacterial infections. Apart from coptisin, there are also palmatine, epiberberine, and jatrorrhizine. These compounds alongside berberine are known to be effective in modulation of hyperglycemia and hyperlipidemia [33].

Structural analysis conducted by Shang et al. (2020) [33] of berberine, coptisine, palmatine, epiberberine, and jatrorrhizine stated that the methylene-dioxy groups found at C2 and C3 and/or C9 and C10 are functional groups that are key in the antihyperglycemic and antihyperlipidemic effects. Furthermore, the oxidized form of the methylene-dioxy group in epiberberine compounds exerts inhibitory activity on aldose reductase. Aldose reductase (AR) is the first enzyme in the polyol formation pathway from glucose with NADPH as a cofactor [37]. In hyperglycemia, more glucose is converted to sorbitol by AR, resulting in the accumulation of ROS and depletion of nicotinamide adenine dinucleotide phosphate (NADPH) and glutathione (GSH) in various tissues, including heart tissue, blood vessels, neurons, eyes, and kidneys [38].

The results of the structure-activity relationship (SAR) analysis that has been carried out by Shang et al. (2020) [33] show that berberine and coptisine have better anti-hyperglycemic effects compared to the other three compounds. This can be explained where berberine and coptisine have methylene-dioxy groups at C2 and C3 and/or C9 and C10, while the three left compounds have substituted by methoxy or phenolic at C2 and C3 positions. These results are shown in Figure 3 and Table 2 below. Research conducted by Kou et al. (2016) [39] stated that the combination of these five compounds together showed synergy in reducing cholesterol levels in HepG2 cells and hamsters with hypercholesterolemic conditions.

## 4. Berberine as an Antidiabetic Compound

Several studies have reviewed the activity of berberine as an antidiabetic compound [19,27,28,29,33]. As an antidiabetic compound, berberine promotes glycolysis through increased glucokinase activity, increased insulin secretion, and suppressed hepatic gluconeogenesis and adipogenesis. This mechanism mainly occurs via AMPK. Other studies have proposed the mechanism of antidiabetic action of berberine through other pathways/enzymes [19,30]. A brief overview of the mechanism of action of berberine can be seen in Figure 4, which shows the activity of berberine as an antidiabetic compound with other supporting activities such as antioxidant and anti-inflammation.

### 4.1. Berberine as Anti-Diabetic Agent via the AMPK Pathway

AMPK is an anergy-sensing/signaling system that is activated/inactivated based on the energy level in the cell, namely the AMP/ATP ratio. AMPK activation increases insulin sensitivity and regulates mitochondrial function [19,40,41]. Berberine activates AMPK by inducing phosphorylation of Thr172 on the α subunit of AMPK [42,43,44]. Berberine can also activate AMPK by increasing the AMP/ATP ratio by inhibiting ATP biosynthesis in mitochondria [45,46]. In mitochondria, berberine directly inhibits the monoamine oxidase (MAO) enzyme on the mitochondrial outer membrane [47,48,49].

Several studies have shown that berberine inhibits mitochondrial function associated with AMPK activation [46]. Berberine is known to have inhibitory activity on complex I of the electron transport chain responsible for mitochondrial respiration. The common antidiabetic drugs metformin and rosiglitazone also share this inhibitory activity. Inhibition of complex I causes mitochondrial function in ATP biosynthesis to be inhibited, increasing the AMP/ATP ratio, which then causes AMPK to become active. This inhibition of mitochondrial function is one of the essential keys in the activity of berberine to prevent liver fat, reduce blood glucose levels, and reduce fatty acids [19,50,51].

### 4.2. Berberine as an Anti-Hyperglycemic Agent

Berberine affects glucose metabolism by stimulating glycolysis via increased glucokinase activity, increased insulin secretion, and suppression of hepatic gluconeogenesis and adipogenesis, and these activities are centered on AMPK activation [52,53,54]. In the insulin resistance state, there is a disruption in the Protein Kinase B (Akt)/Phosphoinositide 3-kinase (PI3K)/Insulin receptor substrate 1 (IRS-1) signaling pathway as an insulin response pathway. Berberine via AMPK shows activity in increasing Akt phosphorylation. Hence, Akt becomes active and can stimulate the regulation of expression and translocation of GLUT4 (Glucose transporter type-4) to the plasma membrane. Hence, glucose can be absorbed and metabolized in the cell. In this case, berberine can increase glucose uptake in an insulin resistance cell [19]. Kaboli et al. (2018) [55], in their in silico docking simulation study, tried to target berberine directly to Akt and obtained a binding energy value of berberine to Akt of −7.78 kcal/mol. Unfortunately, this study did not perform docking simulation with native ligand as the standard, so it is not possible to compare the binding energy value with other compounds. These interactions of berberine with Akt by molecular docking simulation can be seen in Table 3, along with other interactions of berberine with several protein targets.

In terms of insulin secretion, berberine is known to increase and activate the incretin hormone glucagon-like peptide-1 (GLP-1). GLP-1 is an important component involved in pancreatic cell survival. GLP-1 activates adenylate cyclase which converts ATP into cyclic adenosine monophosphate (cAMP), which activates the epac protein, causing an increase in intracellular Ca^2+^ concentration. This stimulates the migration and exocytosis of insulin granules. In addition, cAMP is also known to influence insulin granule exocytosis via PKA signaling [19,61,62].

To prove this, Yu et al. (2010) [63] have researched the effect of berberine addition on increasing GLP-1 levels and the relationship between the increase between GLP-1 levels and increased insulin levels. The results showed that berberine could increase GLP-1 levels at both the medium (secretion results) and cellular levels with the highest results at high berberine concentrations (120 mg/kg). It is also seen that the increase in GLP-1 levels is in line with the increase in insulin levels. Thus, it has been proven that berberine shows activity in increasing insulin secretion via increasing GLP-1 levels. Further research conducted by Yu et al. in 2015 [64] showed that berberine could increase GLP-1 secretion via activation of bitter taste receptor (TAS2R) and PLC (phospholipase C)-dependent behavior. However, the molecular mechanism of this interaction has yet to be further studied.

Berberine is also known to improve glucose metabolism by suppressing gluconeogenesis by inhibiting several transcription factors such as forkhead transcription factor O1 (FOXO1), hepatic nuclear factor 4 (HNF-4), and peroxisome proliferator-activated receptor-γ coactivator-1α (PGC-1α). Inhibition of these transcription factors leads to inhibition/termination of the expression of key enzymes of the gluconeogenesis pathway, namely phosphoenolpyruvate carboxykinase (PEPCK) and glucose-6-phosphatase (G6Pase) [65,66]. Gluconeogenesis itself is the process of glucose formation from sources other than carbohydrates, which is something that diabetics patients want to avoid in order to avoid excessive glucose accumulation. Research conducted by Xia et al. (2011) [67] showed that berberine limits the expression of gluconeogenesis enzymes via inhibition of the FOXO1 transcription factor in both fasting and non-fasting states.

### 4.3. Berberine as Anti-Adipogenic and Anti-Hyperlipidemic Agent

Adipogenesis is the process of adipose/fat tissue growth and development, including adipose tissue cell proliferation and differentiation, where excessive fat accumulation can cause insulin resistance. This can occur because the accumulation of fat tissue will cause a state of excess free fatty acids and increase the transfer of free fatty acids to the liver. This can trigger the secretion of pro-inflammatory cytokines such as TNF-α, interleukin (IL)-6, interleukin (IL)-1, interferon gamma (IFNγ), and monocyte chemoattractant protein-1 (MCP-1) by visceral fat through the portal vein, which can then cause insulin resistance. One example of a pro-inflammatory cytokine, inducible nitric oxide synthase (iNOS), can lead to increased production of nitric oxide (·NO) and peroxynitrite derivatives (ONOO^−^), which can cause nitration of IRS-1/PI3K/Akt, thereby disrupting signaling pathways in GLUT4 translocation to the cell membrane. From this, it can be seen how fat accumulation can cause a state of insulin resistance. Berberine is known to decrease the expression of adipogenesis enzymes. Thus, the decreased expression of adipogenesis enzymes by berberine causes suppression of the fatty acid biosynthesis pathway, one of the causes of insulin resistance [19,68].

Berberine is known to have acted as an anti-adipogenic compound by suppressing the regulation of several transcription factors related to adipogenesis such as the peroxisome proliferator-activated receptor gamma (PPARG), CCAAT enhancer binding protein α (C/EBPα), and sterol response element binding protein-1c (SREBP-1c) [69]. PPARG is a regulator of many fat-specific genes and is the main regulator that drives adipogenesis. Meanwhile, C/EBPα and SREBP-1c are transcription factors that play a role in the expression of adipogenesis enzymes such as fatty acid synthetase (FAS), acetyl-CoA carboxylase (ACC), and acetyl-CoA synthetase (ACS).

Huang and Liu (2022) have conducted research on the interaction of berberine with PPARG through an in silico method, which is molecular docking simulations. The docking simulation results showed that berberine has hydrophobic interactions with residues Ile281, Arg288, and Ile341 of PPARG, which can be seen in Table 3. The results also showed that berberine has a higher affinity value compared to the native ligand of the PPARG complex (PDB ID:1I7I), with a binding energy value of berberine-PPARG of −8.4 kcal/mol and a binding energy value of ligand-PPARG of −7.6 kcal/mol. These results indicate that berberine compounds have binding potential toward PPARG [57].

### 4.4. Berberine as Anti-Inflammatory and Antioxidant Agent

Besides showing activity as an antidiabetic compound by improving insulin sensitivity and secretion and suppressing gluconeogenesis and adipogenesis pathways, berberine exhibits other bioactive activities such as having anti-inflammatory and antioxidant traits. As previously described, pro-inflammatory cytokines can cause insulin resistance. In addition, berberine is also known to act as an anti-inflammatory compound by inhibiting the transcription factor Nf-κB (nuclear factor kappa B) which can encourage the expression of pro-inflammatory cytokines such as TNF-α, cyclooxygenase-2 (COX2), and iNOS. The inhibitory activity of berberine against Nf-κB is known by suppressing the phosphorylation of IKK-β at Ser181 residue. Hence, IKK-β becomes stable, and Nf-κB is inhibited, as IKK-β proteins mask the nuclear localization signals of NF-κB and keep NF-κB in an inactive state [70]. Li et al. (2023) [58] have conducted a docking simulation that shows the orientation of berberine within IKK-β. Unfortunately, this study did not detail the interactions that occur. The anti-inflammatory effect of berberine is also likely to be mediated through AMPK due to the experiments that have been conducted. It was seen that the addition of AMPK inhibitors decreased the inhibitory effect of berberine on the production of pro-inflammatory cytokines [19].

Apart from being anti-inflammatory, berberine also has another advantage as an antioxidant compound. Antioxidants are needed as antidotes to free radicals, which, when accumulated in large quantities, can cause oxidative stress. Oxidative stress is a condition caused by an imbalance between free radical formation and free radical scavenging, such as increased free radical production, reduced antioxidant defense activity, or both. Free radicals can be formed from an incomplete oxidative phosphorylation process. Oxidative stress in patients with hyperglycemia can lead to the development of various diabetic complications, including nephropathy, retinopathy, and neuropathy [1,71,72]. 

The inhibitory activity of berberine on complex I of the respiration chain is one of the activities of berberine as an antioxidant, where complex I of the mitochondrial respiration chain is a major contributor to superoxide production [50,51,73]. In addition, berberine is also known to activate the translocation of Nrf2 (nuclear factor erythroid 2-related factor 2), which is a transcription factor of superoxide dismutase (SOD) and glutathione (GSH), thus increasing the mRNA expression of SOD, GSH, and GSH-peroxidase glutathione which acts as an antioxidant substance in the body to counteract free radicals by converting superoxide radicals (O_2_^−^) into oxygen (O_2_) and water (H_2_O) [74,75]. Berberine is also said to regulate uncoupling protein 2 (UCP2), where UCP2 is a mitochondrial inner membrane protein that is negatively associated with ROS production and oxidative stress. Berberine treatment associated with the upregulation of UCP2 showed reduced atherosclerosis in mice and attenuated non-alcoholic fatty liver disease in rats [19].

## 5. Development of Berberine Usage

Berberine has a long history as a traditional Chinese medicine commonly used to treat various diseases, and recent studies have shown that berberine has various benefits in the treatment of various health problems [76]. Several toxicity tests and clinical trials have been conducted on berberine compounds.

Acute toxicity tests aim to detect the intrinsic toxicity of a substance and obtain information on the LD_50_ value. The LD_50_ value indicates the dose of test substance that then causes 50% death in test animals acutely [77]. In addition to toxicity testing, determining drug bioavailability is an important parameter to determine the amount and speed of drugs absorbed in the body. It can be determined by measuring drug levels in the blood against time or from its excretion in the urine [29,78].

Acute toxicity tests on rat samples have been conducted by Kheir et al. (2010) [79] through three routes, including intravenous injection, intraperitoneal injection, and oral administration via intragastric. From the results of the study, it is known that the bioavailability of berberine from highest to lowest in order is the intravenous injection, intraperitoneal injection, and lastly, oral intragastric. In this study, it was not possible to determine the exact LD_50_ value for oral administration due to the absorption limit of berberine via the oral route. However, it is known that the safe dose of berberine for oral consumption in rats is 20.8 g/kg body weight, and the toxic dose is 41.6 g/kg body weight. Meanwhile, the safe dose of berberine for human consumption is 1/7 of the rat dose, which is 2.97 g/kg body weight [79].

Yi et al. (2013) [80] in their study showed that berberine isolated from *Rhizoma coptidis* has an LD_50_ value of 713.57 mg/kg and is classified as toxicity category III. However, a recent study conducted by Zhang et al. (2022) [81] stated that berberine in salt form (berberine hydrochloride) has a low toxicity value (toxicity category V) with an LD_50_ value of 2.83 ± 1.21 g/kg. Table 4 below summarizes the LD_50_ values of berberine compounds in both their salt and extract forms from several references.

A study conducted by Zhang et al. (2008) [83] on 116 people with diabetes showed that consumption of 1 g of berberine per day for 3 months can reduce fasting blood sugar levels by 20%. The same study also showed that the administration of berberine can reduce the levels of aggregated hemoglobin (HbA1c) by 12% and improve blood lipid conditions such as cholesterol and triglycerides [83]. Berberine is said to be potentially effective as other oral antidiabetic drugs such as metformin, glipizide, and rosiglitazone [25].

Unlike common antidiabetic drugs that can have side effects such as weight gain, several studies have shown that berberine is effective in reducing weight gain in patients with type 2 diabetes [24,84]. In a 12-week study conducted by Hu et al. (2012) [26] of people with a diagnosis of obesity, consumption of 500 mg berberine three times a day can reduce body weight by an average of 5 pounds (about 2.3 kg). It was recorded that participants reduced up to 3.6% of body fat levels [26]. Besides that, several pre-clinical trials and clinical trials of berberine compounds have also shown good effects of berberine on cardiovascular, liver, and kidney diseases [19,81]. Meanwhile, the side effects of berberine tested in one of the short-term clinical trials showed that berberine only caused minor gastrointestinal symptoms such as constipation, flatulence, and diarrhea [27].

There are still some drawbacks of berberine that have been known. As mentioned before, it was hard to determine the LD_50_ value for oral administration of berberine due to the absorption limit of berberine via the oral route. Some studies said that the bioavailability of berberine is less than 1%, and about half of the dose is not absorbed by the intestine [56,85]. Berberine in its salt form may help with bioavailability and toxicity levels issues [81]. Another way is the oxidized form of berberine, namely oxyberberine. Oxyberberine could be formed from berberine via an oxidation reaction by intestinal microflora [56]. Li et al. (2019) found that oxyberberine exhibits a more favorable safety profile compared to berberine, with an LD_50_ value above 5000 mg/kg in mice [86]. Other than that, oxyberberine showed superior antidiabetic effect to metformin. It also effectively inhibits inflammation response and exhibits more anti-inflammatory properties than berberine [56,86]. This encourages the importance of studying berberine and its derivatives as antidiabetic agents that have the potential as alternative diabetes drugs.

## 6. Conclusions

Diabetes mellitus is a metabolic disorder that causes hyperglycemia, leading to various chronic complications and death. The treatment of type 2 DM caused by insulin resistance involves antidiabetic drugs that have a high level of risk. Therefore, the exploration of natural compounds as alternative antidiabetic drugs has been carried out, one of which is berberine.

Berberine is an isoquinoline alkaloid compound that shows activity as an anti-hyperglycemia agent by improving the regulation of glucose and lipid metabolism in the body. Berberine can inhibit mitochondrial function and activate the AMPK pathway, which is then involved with several other pathways related to glucose or fat metabolism. Apart from being anti-diabetic, berberine is also known to act as an antioxidant by reducing ROS accumulation and can act as an anti-inflammatory compound.

Many mechanisms of berberine have yet to be uncovered. However, from the mechanisms that have been proposed, berberine shows great potential as an oral antidiabetic drug. However, berberine has some drawbacks, such as low absorption and high level of toxicity of berberine in its free form. Future studies of berberine are needed, especially in the exploration of berberine derivatives or modification of the berberine structure, so that a berberine compound that has not only good antidiabetic potential but is also such that good pharmacokinetics can be obtained.

## Figures and Tables

**Figure 1 biology-12-00973-f001:**
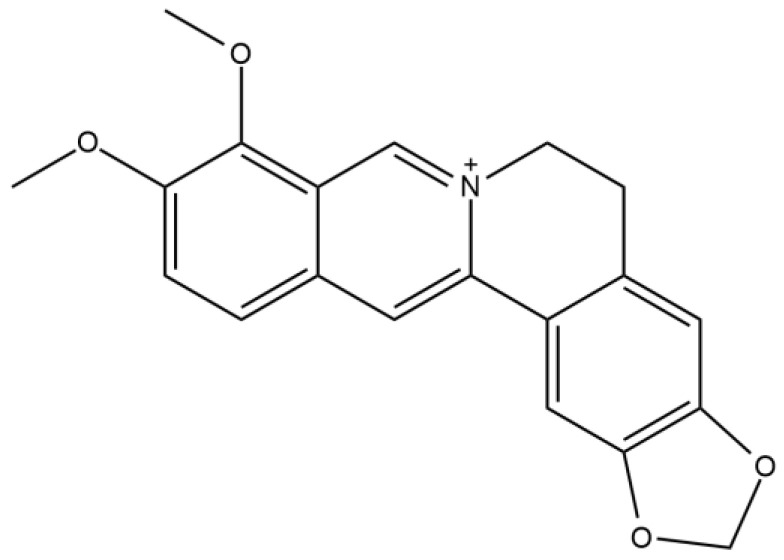
Berberine structure. Quartenery base structure of isoquinoline alkaloid berberine that have been used as traditional medicine in various regions and known to have antidiabetic activity.

**Figure 2 biology-12-00973-f002:**
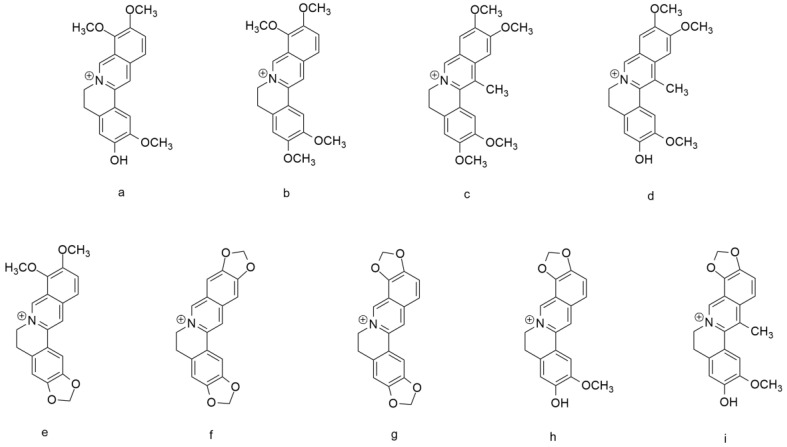
Berberine and several alkaloid compounds. (**a**) jatrorrhizine, (**b**) palmatine, (**c**) pseudodehydrocorydaline, (**d**) dehydrocorylbulbine, (**e**) berberine, (**f**) pesudocoptisine, (**g**) coptisine, (**h**) dehydroisoapocavadine, and (**i**) dehydrocheilanthifoline.

**Figure 3 biology-12-00973-f003:**
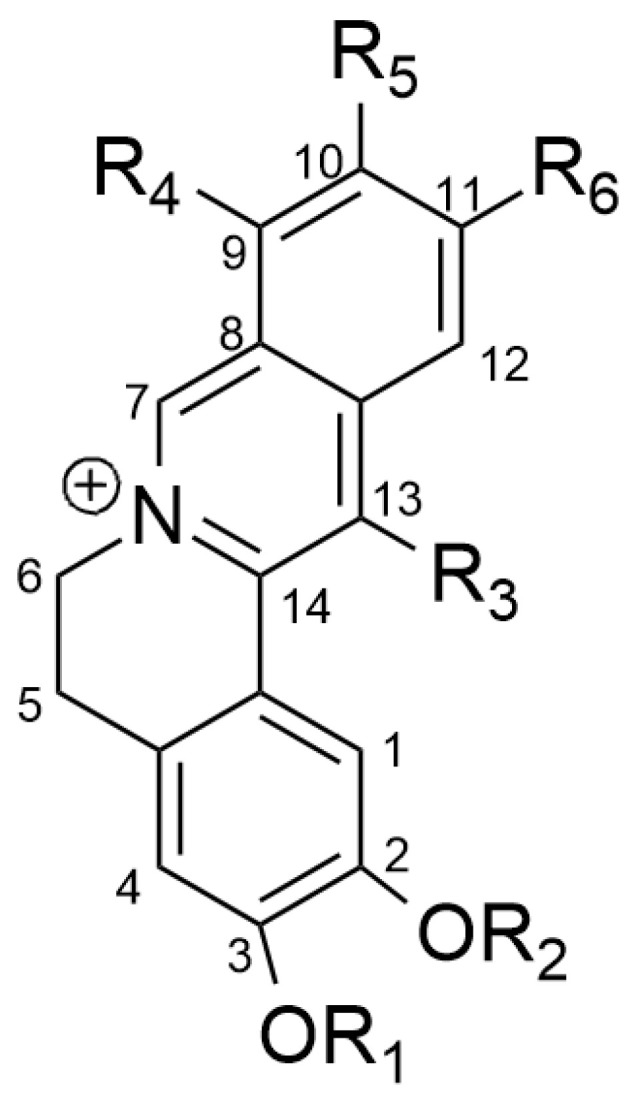
SAR analysis of berberine, coptisine, palmatine, epiberberine, and jatrorrhizine.

**Figure 4 biology-12-00973-f004:**
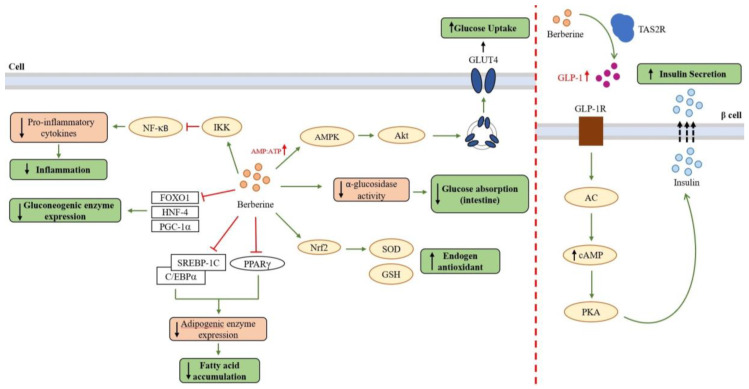
A brief overview of the mechanism of action of berberine through several pathways as an antidiabetic compound with other support activities such as anti-inflammation and antioxidant [19].

**Table 1 biology-12-00973-t001:** Neuraminidase inhibition activity of berberine and several alkaloid compounds isolated from the roots of *Corydalis turtschaninovii* with its IC_50_ value [36].

Compounds	IC_50_ (μM)
Berberine	13.5 ± 2.3
Jatrorrhizine	37.0 ± 1.8
Pseudocoptisine	65.2 ± 4.5
Coptisine	25.1 ± 0.8
Palmatine	12.8 ± 1.5
Pseudodehydrocorydaline	32.6 ± 2.1
Dehydrocorylbulbine	41.3 ± 3.5

**Table 2 biology-12-00973-t002:** SAR analysis of berberine (1), coptisine (2), palamatine (3), epiberberine (4), and jatrorrhizine (5).

	R_1_	R_2_	R_3_	R_4_	R_5_	R_6_
1	-CH_2_-		H	OMe	OMe	H
2	-CH_2_-		H	-OCH_2_O-		H
3	Me	Me	H	OMe	OMe	H
4	Me	Me	H	-OCH_2_O-		H
5	Me	H	H	OMe	OMe	H

The methylene-dioxy groups at C2, C3, C9, and C10 showed anti hyperglycemic and antihyperlipidemic effects. These groups are shown in the compounds berberine (1) and coptisine (2) and explained why these two compounds have good antihyperglycemic effects. Meanwhile, the oxidized form of methylene-dioxy group (-OCH2O-) on epiberberine (4) exerts inhibitory activity on aldose reductase (AR) [27].

**Table 3 biology-12-00973-t003:** In silico studies of berberine and its interactions.

Protein Targets	Binding Energy (kcal/mol)	Interactions	References
PI3K Akt GSK-3β Keap-1	−5.8 −8.3 −7.6 −8.6	Gln8, Lys42 Lys179 Lys183 Val97, Ile236	[56]
Akt	−7.83	Arg4, Thr291, Val164, Met281, Ala177, Leu156, Met227, Asp292	[55]
PPARG	−8.4	Ile281, Arg288, Ile341	[57]
FOXO3 FOXO4 IKK-β	−8.9 −9.0 −7.7	*n/i *n/i *n/i	[58]
α-glucosidase DPP-IV Insulin receptor (IR)	−7.9 −8.9 −6.3	Asp568, Tyr709 Tyr547, Gln553, Ser630 Gln1004	[59]
TNF-α IL-6	−6.24 −4.94	Lys65, Phe144 Asn135	[60]

PI3K, phosphoinositide 3-kinase; Akt, protein kinase B; GSK-3β, glycogen synthase kinase 3 beta; Keap-1, Kelch-like ECH-associated protein 1; PPARG, peroxisome proliferator-activated receptor gamma; FOXO3, forkhead box O3; FOXO4, forkhead box O4; IKKβ, IκB kinase beta; DPP-IV, dipeptidyl peptidase-IV; IR, insulin receptor; TNF-α, tumor necrosis factor alpha; IL-6, interleukin 6. *n/i: no information.

**Table 4 biology-12-00973-t004:** LD_50_ value of berberine via IV, IP, and IG administration.

Compound	LD_50_			References
	IV	IP	IG	
Berberine hydrochloride	9.04 ± 0.8 mg/kg	57.61 ± 23.32 mg/kg	LD_50_ not found; 20.8 g/kg (safe dose) 41.6 g/kg (toxic dose)	[79]
	-	-	2.83 ± 1.21 g/kg	[81]
Berberine from *Rhizoma coptidis*	-	-	713.57 mg/kg (0.713 g/kg)	[80]
Powdered root *B. vulgaris*	-	-	2.60 g/kg	[82]
Extract of *B. vulgaris*	-	-	1.28 g/kg	[82]
Berberine sulphate	-	205 mg/kg	-	[82]
Berberine	-	23 mg/kg	-	[82]

IV, intravena; IP, intraperitoneal; IG, oral intragastric.

## Data Availability

Not applicable.

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
