# Peer review of "Berberine and Its Study as an Antidiabetic Compound"

_biology, 2023, doi:10.3390/biology12070973_

Round 1

Reviewer 1 Report

In this review article, the authors summarized the role of Berberine as an antidiabetic agent with various mechanistic pathways reported in literature. In addition, other therapeutic properties of berberine like antioxidant, anti-inflammatory etc. are also discussed. This will be an important review for berberine and its derivatives in the corresponding topic.  My comments are as follows:

1. The scope and importance of this review article is high due to the following reasons:

 a) The review article emphasized the prevention of T2DM using Berberine which are from various readily available plant sources and are known to act as antioxidants and anti-inflammatory agents.

b) Due to the usage of natural remedies, side effects will be very less as compared to the traditional medications. (toxicities are reported in the form of LD50 values and found to be compatible)

c) The best part of Berberine is that it might prevent T2DM by means of different mechanistic pathways.  

2. The review is well written and presented with rational scientific language with no typographical errors. The title and the abstract represent the objective of the manuscript.

 Section 2 may be included in the introduction part. Sections 3 and 4 gives us a structural insight about the berberine and its derivatives as well as SAR studies with authentic examples.

3. The schemes and images are presented correctly (structures all molecules are correct) and texts presented are in line with them.

4. Section 5 describes the role of berberine to control T2DM in different mechanistic pathways and are represented with proper schematic pathways with authentic references.

5.  The IC50 values about the inhibition studies may be presented in a tabular form. In-sillico study report could be incorporated with a diagram with all possible interactions of berberine with the target.

6. The references are formatted in line with the journal standard.

7. What will be future directions/STUDIES required for berberine to designate as a therapeutic agent for prevention of T2DM?

8. It is suggested to mention the importance of synthetic efforts towards development of antidiabetic medicine. In this regard, it is recommended to emphasis the importance of iminosugars and sugar derivatives as an antidiabetic agent and suggested to cite following relevant articles in the introduction section.

a.      Po-Sen Tseng, Dr. Chennaiah Ande, Prof. Dr. Kelley W. Moremen, Prof. Dr. David Crich. Influence of Side Chain Conformation on the Activity of Glycosidase Inhibitors. Angewandte Chemie International Edition. 2022. (https://doi.org/10.1002/anie.202217809)

b.     Rajasekaran, P.; Ande, C.; Vankar, Y. D. Synthesis of (5,6 & 6,6)-oxa-oxa annulated sugars as glycosidase inhibitors from 2-formyl galactal using iodocyclization as a key step. ARKIVOC 2022, vi, 5−23.

c.      Chennaiah, A.; Bhowmick, S.; Vankar, Y. D. Conversion of glycals into vicinal-1,2-diazides and 1,2-(or 2,1)-azidoacetates using hypervalent iodine reagents and Me3SiN3. Application in the synthesis of N-glycopeptides, pseudo-trisaccharides and an iminosugar. RSC Adv. 2017, 7, 41755−41762.

Author Response

We thank you very much for the careful revision to our manuscript. Please see the attachment.

Reviewer 2 Report

This review provides a comprehensive overview of the recent perspective on berberine, highlighting its roles as an antidiabetic, anti-adipogenic, anti-hyperlipidemic, anti-inflammatory, and antioxidant agent, and elucidating its mechanisms as a multifunctional natural compound. It complements the article titled "A Review of Fibraurea tinctoria and Its Component, Berberine, as an Antidiabetic and Antioxidant," recently published in Molecules. Together, these two articles offer a comprehensive understanding of the beneficial properties of berberine in the treatment of various diseases. A few comments: 1. Add reference to Line 115 (One of those alkaloid compounds named berberine is an isoquinoline alkaloid that acts as a therapeutic agent in the treatment of T2DM). 2. Add reference to Line 116 (Berberine has been used as a traditional medicine in China, India, and the Middle East region for more than 400 years). 3. Line 159, Which five compounds did the authors make reference to? 4. Recommend incorporating the oxidized form of berberine and illustrating its diverse functions. 5. What is the reference of work done by Kou, et al. in line 173? 6. Add the reference to line 186 (Several studies have reviewed the activity of berberine as an antidiabetic compound). 7. Is the mechanism of action of berberine, depicted in Figure 4, widely accepted as an effective pathway for its role as an antidiabetic compound?

Author Response

(The authors gave the same response as above.)
